# Model Selection in Atmospheric Remote Sensing with Application to Aerosol Retrieval from DSCOVR/EPIC. Part 2: Numerical Analysis

**Sruthy Sasi** [1], **Vijay Natraj** [2,*], **Víctor Molina García** [1], **Dmitry S. Efremenko** [1], **Diego Loyola** [1] and **Adrian Doicu** [1]

1  German Aerospace Center (DLR), Remote Sensing Technology Institute, 82234 Oberpfaffenhofen, Germany; Sruthy.Sasi@dlr.de (S.S.); Victor.MolinaGarcia@dlr.de (V.M.G.); dmitry.efremenko@dlr.de (D.S.E.); diego.loyola@dlr.de (D.L.); adrian.doicu@dlr.de (A.D.)
2  Jet Propulsion Laboratory (NASA-JPL), California Institute of Technology, 4800 Oak Grove Drive, Pasadena, CA 91109, USA
*  Correspondence: vijay.natraj@jpl.nasa.gov

**Abstract:** An algorithm for retrieving aerosol parameters by taking into account the uncertainty in aerosol model selection is applied to the retrieval of aerosol optical thickness and aerosol layer height from synthetic measurements from the EPIC sensor onboard the Deep Space Climate Observatory. The synthetic measurements are generated using aerosol models derived from AERONET measurements at different sites, while other commonly used aerosol models, such as OPAC, GOCART, OMI, and MODIS databases are used in the retrieval. The numerical analysis is focused on the estimation of retrieval errors when the true aerosol model is unknown. We found that the best aerosol model is the one with a value of the asymmetry parameter and an angular variation of the phase function around the viewing direction that is close to the values corresponding to the reference aerosol model.

**Keywords:** model selection; retrieval algorithm; DSCOVR/EPIC

## 1. Introduction

The retrieval of aerosol and cloud optical thickness, as well as aerosol layer height and cloud top height, requires the selection of a model that describes their microphysical properties. If there is insufficient information for an appropriate microphysical model selection, the solution accuracy can be improved if this model uncertainty is taken into account and appropriately quantified.

In Ref. [1], we presented a retrieval algorithm that takes into account uncertainty in model selection. The algorithm is based on (i) the iteratively regularized Gauss–Newton method to compute the solution for each model, (ii) a linearization of the forward model around the solution, and (iii) an extension of maximum marginal likelihood estimation and generalized cross-validation methods to model selection and data error variance estimation. Essentially, the algorithm includes four selection models corresponding to (i) the two parameter choice methods used (maximum marginal likelihood estimation and generalized cross-validation), and (ii) the two settings in which the relative evidence is treated (stochastic and deterministic). In practice, the algorithm can be used for the retrieval of (i) aerosol parameters from different spectral instruments, and (ii) cloud parameters, when the selection of the cloud type and/or the effective radius may play an important role.

The above algorithm was applied to the retrieval of aerosol optical thickness and layer height from synthetic measurements corresponding to the EPIC instrument onboard the Deep Space Climate Observatory [2]. Specifically, Channels 7 and 8 in the Oxygen B-band at 680 and 687.75 nm, respectively,

and Channels 9 and 10 in the Oxygen A-band at 764 and 779.5 nm, respectively, were used for this purpose. The simulations utilize aerosol models implemented in the MODIS aerosol algorithm over land [3]; the surface albedo is either assumed to be known or included in the retrieval.

In this paper, we utilize the methodology developed in Ref. [1] to perform retrievals of synthetic measurements from the EPIC instrument. For this purpose,

1.  a set of aerosol models derived from Aerosol Robotic Network (AERONET) measurements at different sites [4,5] is used as reference, under the simplified assumption that the surface albedo is known,
2.  the OPAC [6], GOCART [7], OMI [8], and MODIS [3] aerosol models are employed in the retrieval.

The numerical analysis is devoted to the estimation of errors in the retrieval of aerosol optical thickness and layer height, when the true aerosol model is unknown.

The paper is organized as follows. In Section 2, we recapitulate the main features of the aerosol retrieval algorithm. In Section 3, we describe the aerosol models used in this study. We present the results of our numerical analysis in Section 4.

## 2. Algorithm Description

For $N_{\mathrm{m}}$ microphysical aerosol models, we consider the scaled white-noise data model

$$\overline{\mathbf{y}}^{\delta} = \overline{\mathbf{F}}_m(\mathbf{x}) + \overline{\delta}_m, \tag{1}$$

where $\overline{\mathbf{F}}_m(\mathbf{x}) \in \mathbb{R}^M$ is the forward model corresponding to the model $m$, $m = 1, \ldots, N_{\mathrm{m}}$, $\overline{\mathbf{y}}^{\delta} \in \mathbb{R}^M$ the measurement vector or the noisy data vector, and $\overline{\delta}_m \in \mathbb{R}^M$ the data error vector summing the contributions of the measurement and model error vectors.

In a stochastic setting, $\overline{\delta}_m$ and $\mathbf{x}$ are random vectors, and Equation (1) is solved by means of a Bayesian approach. Specifically, for $\mathbf{x} \sim \mathsf{N}(\mathbf{x}_{\mathrm{a}}, \mathbf{C}_{\mathrm{x}} = \sigma_m^2(\alpha \mathbf{L}^T \mathbf{L})^{-1})$ and $\overline{\delta}_m \sim \mathsf{N}(\mathbf{0}, \overline{\mathbf{C}}_{\delta m} = \sigma_m^2 \mathbf{I}_M)$, the maximum *a posteriori* estimator $\widehat{\mathbf{x}}_{m\alpha}^{\delta}$ is computed as

$$\widehat{\mathbf{x}}_{m\alpha}^{\delta} = \arg \min_{\mathbf{x}} V_{\alpha}(\mathbf{x} \mid \overline{\mathbf{y}}^{\delta}, m), \tag{2}$$

where

$$V_{\alpha}(\mathbf{x} \mid \overline{\mathbf{y}}^{\delta}, m) = \frac{1}{\sigma_m^2} \left[ \left\| \overline{\mathbf{y}}^{\delta} - \overline{\mathbf{F}}_m(\mathbf{x}) \right\|^2 + \alpha \left\| \mathbf{L}(\mathbf{x} - \mathbf{x}_{\mathrm{a}}) \right\|^2 \right]$$

is the *a posteriori* potential, the notation $\mathsf{N}(\mathbf{x}_{\mathrm{mean}}, \mathbf{C}_{\mathrm{x}})$ stands for a normal distribution with mean $\mathbf{x}_{\mathrm{mean}}$ and covariance matrix $\mathbf{C}_{\mathrm{x}}$, $\mathbf{x}_{\mathrm{a}}$ is the a priori state vector, $\alpha = \sigma_m^2 / \sigma_{\mathrm{x}}^2$ is the regularization parameter, i.e., the ratio of data error variance $\sigma_m^2$ and a priori state variance $\sigma_{\mathrm{x}}^2$, and $\mathbf{L}$ is the regularization matrix.

In a deterministic setting, $\overline{\delta}_m$ is characterized by the noise level $\Delta_m$ (defined as an upper bound for $\overline{\delta}_m$ i.e., $\|\overline{\delta}_m\| \leq \Delta_m$), $\mathbf{x}$ is a deterministic vector, and we are faced with the solution of the nonlinear equation $\overline{\mathbf{y}}^{\delta} = \overline{\mathbf{F}}_m(\mathbf{x})$. In the framework of Tikhonov regularization, a regularized solution $\mathbf{x}_{m\alpha}^{\delta}$ to the nonlinear equation $\overline{\mathbf{y}}^{\delta} = \overline{\mathbf{F}}_m(\mathbf{x})$ minimizes the Tikhonov function $\mathcal{F}_{m\alpha}(\mathbf{x}) = \sigma_m^2 V_{\alpha}(\mathbf{x} \mid \overline{\mathbf{y}}^{\delta}, m)$; hence, the maximum a posteriori estimate coincides with the Tikhonov solution, i.e., $\widehat{\mathbf{x}}_{m\alpha}^{\delta} = \mathbf{x}_{m\alpha}^{\delta}$. Since the computation of the regularized solution $\mathbf{x}_{m\alpha}^{\delta}$ in the framework of the method of Tikhonov regularization requires knowledge of the optimal value of the regularization parameter $\widehat{\alpha}$, the nonlinear equation $\overline{\mathbf{y}}^{\delta} = \overline{\mathbf{F}}_m(\mathbf{x})$ is solved by means of the iteratively regularized Gauss–Newton method. In addition to the optimal value of the regularization parameter $\widehat{\alpha}$, this method also provides the corresponding regularized solution $\mathbf{x}_{m\widehat{\alpha}}^{\delta}$.

The key quantity in Bayesian model selection is the relative evidence $p(m \mid \overline{\mathbf{y}}^\delta)$, also known as the a posteriori probability of the model $m$ given the measurement $\overline{\mathbf{y}}^\delta$. In terms of this quantity, the mean and maximum solution estimates are defined by

$$\widehat{\mathbf{x}}^\delta_{\text{mean}} = \sum_{m=1}^{N_{\text{m}}} \mathbf{x}^\delta_{m\widehat{\alpha}} p(m \mid \overline{\mathbf{y}}^\delta), \tag{3}$$

and

$$\widehat{\mathbf{x}}^\delta_{\text{max}} = \mathbf{x}^\delta_{m^\star \widehat{\alpha}}, \quad m^\star = \arg \max_m p(m \mid \overline{\mathbf{y}}^\delta), \tag{4}$$

respectively. The relative evidence $p(m \mid \overline{\mathbf{y}}^\delta)$ can be computed in a stochastic or a deterministic setting.

1. In a stochastic setting, $p(m \mid \overline{\mathbf{y}}^\delta)$ is calculated using the relation

$$p(m \mid \overline{\mathbf{y}}^\delta) = \frac{p(\overline{\mathbf{y}}^\delta \mid m)}{\sum_{m=1}^{N_{\text{m}}} p(\overline{\mathbf{y}}^\delta \mid m)},$$

   where $p(\overline{\mathbf{y}}^\delta \mid m)$ is the marginal likelihood density. By assuming a linearization of the forward model around the solution, the marginal likelihood density $p(\overline{\mathbf{y}}^\delta \mid m)$ can be computed analytically provided that the data error variance $\sigma^2_m$ is known. Estimates for $\sigma^2_m$ can be obtained in the framework of maximum marginal likelihood estimation [9–11] and generalized cross-validation methods [12,13].

2. In a deterministic setting, $p(m \mid \overline{\mathbf{y}}^\delta)$ is regarded as a merit function characterizing the solution $\mathbf{x}^\delta_{m\widehat{\alpha}}$. More precisely, $p(m \mid \overline{\mathbf{y}}^\delta)$ is defined in terms of the marginal likelihood function or the generalized cross-validation function. In the latter case, we have

$$p(m \mid \mathbf{y}^\delta) = \frac{1/v(m)}{\sum_{m=1}^{N_{\text{m}}} 1/v(m)}. \tag{5}$$

   where

$$v(m) = \frac{||\mathbf{r}^\delta_{m\widehat{\alpha}}||^2}{[\text{trace}(\mathbf{I} - \widehat{\mathbf{A}}_{m\widehat{\alpha}})]^2}, \tag{6}$$

   is the generalized cross-validation function, $\mathbf{r}^\delta_{m\widehat{\alpha}} = \overline{\mathbf{y}}^\delta - \overline{\mathbf{F}}_m(\mathbf{x}^\delta_{m\widehat{\alpha}})$ is the nonlinear residual vector, $\widehat{\mathbf{A}}_{m\widehat{\alpha}} = \overline{\mathbf{K}}_{m\widehat{\alpha}} \overline{\mathbf{K}}^\dagger_{m\widehat{\alpha}}$ is the influence matrix, $\overline{\mathbf{K}}_{m\widehat{\alpha}}$ is the Jacobian matrix, and $\overline{\mathbf{K}}^\dagger_{m\widehat{\alpha}}$ is the generalized inverse at the solution $\mathbf{x}^\delta_{m\widehat{\alpha}}$.

   The numerical simulations performed in Ref. [1] show that

1. the differences between the results corresponding to the stochastic and deterministic settings are not significant, and

2. the maximum solution estimate $\widehat{\mathbf{x}}^\delta_{\text{max}}$ is completely unrealistic.

   In the present study, we assume a deterministic setting and compute the mean solution estimate $\widehat{\mathbf{x}}^\delta_{\text{mean}}$ and the maximum solution estimate $\widehat{\mathbf{x}}^\delta_{\text{max}}$ by means of Equations (3) and (4), respectively, and the relative evidence $p(m \mid \overline{\mathbf{y}}^\delta)$ according to Equations (5) and (6).

## 3. Aerosol Models

To describe the wide range of possible compositions, the aerosol particles are modeled as components, each of them representing an internal mixture of all chemical substances that have a similar origin. The size distribution of an aerosol component is assumed to be log-normal, described by the number size distribution

$$\frac{\mathrm{d}N(r)}{\mathrm{d}\ln r} = \frac{N_0}{\sqrt{2\pi}\sigma} \exp\left[-\frac{(\ln r - \ln r_{\text{mod}})^2}{2\sigma^2}\right], \tag{7}$$

where $r_{\mathrm{mod}}$ is the modal or median radius of the number size distribution, $\sigma$ is the standard deviation, and

$$N_0 = \int_0^\infty \frac{\mathrm{d}N(r)}{\mathrm{d}\ln r}\,\mathrm{d}\ln r \tag{8}$$

is the total number of particles (per cross-section of atmospheric column). Alternatively, the log-normal model can be described by the volume size distribution

$$\frac{\mathrm{d}V(r)}{\mathrm{d}\ln r} = \frac{V_0}{\sqrt{2\pi}\sigma}\exp\left[-\frac{(\ln r - \ln r_{\mathrm{v}})^2}{2\sigma^2}\right], \tag{9}$$

where

$$r_{\mathrm{v}} = r_{\mathrm{mod}}\exp(-3\sigma^2) \tag{10}$$

is the median radius of the volume size distribution and

$$V_0 = \int_0^\infty \frac{4\pi r^3}{3}\frac{\mathrm{d}N(r)}{\mathrm{d}\ln r}\,\mathrm{d}\ln r = N_0\frac{4\pi r_{\mathrm{mod}}^3}{3}\exp(4.5\sigma^2) \tag{11}$$

is the volume of particles (per cross section of atmospheric column). Thus, the size distribution of an aerosol component is characterized by (i) the modal radius $r_{\mathrm{mod}}$, (ii) the standard deviation $\sigma$, and (iii) the total number of particles $N_0$. Alternatively, in addition to the standard deviation $\sigma$, the median radius of the volume size distribution $r_{\mathrm{v}}$ and the volume of particles $V_0$ can be used to characterize the size distribution. When these parameters together with the wavelength-dependent refractive index $m_{\mathrm{aer}}$ are specified, the scattering characteristics of an aerosol component, i.e.,

1. the size averaged extinction and scattering cross sections $\overline{C}_{\mathrm{ext}}$ and $\overline{C}_{\mathrm{sct}}$,
2. the single scattering albedo $\overline{\omega}$, and
3. the coefficients $\overline{a}_n$ of the expansion of the size averaged phase function $\overline{P}(\Theta)$ in terms of Legendre polynomials $P_n(\cos\Theta)$, i.e.

$$\overline{P}(\Theta) = \sum_{n \geq 0} \overline{a}_n P_n(\cos\Theta)$$

can be computed using an electromagnetic scattering model. In particular, for spherical particles, the size averaged quantities are calculated using the formulas

$$\overline{C}_{\mathrm{ext}} = \int_{r_{\mathrm{min}}}^{r_{\mathrm{max}}} C_{\mathrm{ext}}(r)p(r)\,\mathrm{d}r, \tag{12}$$

$$\overline{C}_{\mathrm{sct}} = \int_{r_{\mathrm{min}}}^{r_{\mathrm{max}}} C_{\mathrm{sct}}(r)p(r)\,\mathrm{d}r, \tag{13}$$

$$\overline{\omega} = \frac{\overline{C}_{\mathrm{sct}}}{\overline{C}_{\mathrm{ext}}}, \tag{14}$$

$$\overline{a}_n = \int_{r_{\mathrm{min}}}^{r_{\mathrm{max}}} a_n(r)p(r)\,\mathrm{d}r, \ \ n \geq 0, \tag{15}$$

where $p(r)$ is the probability density function associated with the number size distribution,

$$p(r) = \frac{1}{N_0}\frac{\mathrm{d}N(r)}{\mathrm{d}r}, \tag{16}$$

for which we have

$$p(r)\,\mathrm{d}r = \frac{1}{N_0}\,\mathrm{d}N(r) = \frac{1}{N_0}\frac{\mathrm{d}N(r)}{\mathrm{d}\ln r}\,\mathrm{d}\ln r, \tag{17}$$

$C_{\mathrm{ext}}(r)$, $C_{\mathrm{sct}}(r)$, and $a_n(r)$ correspond to a spherical particle of radius $r$, and $r_{\mathrm{min}}$ and $r_{\mathrm{max}}$ are the lower and upper limits of the size distribution.

The aerosol components can be externally mixed to form aerosol models (classes). External mixture means that there is no physical or chemical interaction between particles of different components. If an aerosol model $m$ consists of $N$ aerosol components, and $\overline{C}_{\text{ext}i}$, $\overline{C}_{\text{sct}i}$, and $\overline{a}_{ni}$ correspond to the $i$th aerosol component, the extinction cross section $\overline{C}_{\text{ext}m}$, the scattering cross section $\overline{C}_{\text{sct}m}$, the single scattering albedo $\overline{\omega}_m$, and the expansion coefficients $\overline{a}_{mn}$ of the aerosol model are computed using the external mixing formulas

$$\overline{C}_{\text{ext}m} = \sum_{i=1}^{N} w_i \overline{C}_{\text{ext}i}, \tag{18}$$

$$\overline{C}_{\text{sct}m} = \sum_{i=1}^{N} w_i \overline{C}_{\text{sct}i}, \tag{19}$$

$$\overline{\omega}_m = \frac{\overline{C}_{\text{sct}m}}{\overline{C}_{\text{ext}m}}, \tag{20}$$

$$\overline{C}_{\text{sct}m} \overline{a}_{mn} = \sum_{i=1}^{N} w_i \overline{C}_{\text{sct}i} \overline{a}_{ni}, \quad n \geq 0, \tag{21}$$

where the weight

$$w_i = \frac{N_{0i}}{\sum_{i=1}^{N} N_{0i}} \tag{22}$$

is the number mixing ratio, and $N_{0i}$ is the total number of particles of the $i$th aerosol component.

For the extinction coefficient, we have the computational formula

$$\sigma_{\text{ext}m} = n_0 \overline{C}_{\text{ext}m}, \tag{23}$$

where

$$n_0 = \frac{\sum_{i=1}^{N} N_{0i}}{V} = \sum_{i=1}^{N} \frac{N_{0i}}{V} = \sum_{i=1}^{N} n_{0i}, \tag{24}$$

is the total number density of the aerosol particles, and $n_{0i}$ is the number density of the $i$th aerosol component. Note that relation (22) can be expressed in terms of $n_{0i}$ as

$$w_i = \frac{N_{0i}}{\sum_{i=1}^{N} N_{0i}} = \frac{n_{0i}}{n_0}. \tag{25}$$

### 3.1. Aerosol Model Sets

This section briefly describes the aerosol model sets used in this study. The parameters of the particle size distributions (i.e., modal radii and standard deviations) and the refractive index data for each set are summarized in Tables A1–A5 in Appendix A.

### 3.1.1. Set 1

The first model set includes the aerosol components from the Optical Properties of Aerosols and Clouds (OPAC) database [6]. These are

1.  the *water-insoluble* part of aerosol particles, consisting mostly of soil particles with a certain amount of organic material;
2.  the *water-soluble* part of aerosols, originating from gas to particle conversion and consisting of various kinds of sulfates, nitrates, and organic, water-soluble substances;
3.  the *soot* component, describing the absorbing black carbon (note that carbon is not soluble in water and therefore, the particles are assumed not to grow with increasing relative humidity);

4. *sea-salt (accumulated and coarse)* particles, consisting of the various kinds of salt contained in seawater;
5. *mineral aerosol (accumulated and coarse)* or desert dust, consisting of a mixture of quartz and clay minerals;
6. *mineral transported*, describing the desert dust that is transported over long distances with a reduced amount of large particles; and
7. the *sulfate* component, describing the amount of sulfate found in the Antarctic aerosol (also used as the stratospheric background aerosol).

Assuming that the aerosol particles are spherical, we generate a database using Mie scattering theory. Essentially, the database gives the values of $\overline{C}_{ext}$, $\overline{C}_{sct}$, $\overline{\omega}$, and $\overline{a}_n$, for (i) each aerosol component, (ii) a set of wavelengths (61 values in the range $0.250\,\mu\text{m} - 40\,\mu\text{m}$), and (iii) a set of relative humidities (8 values: 0.00, 0.50, 0.70, 0.80, 0.90, 0.95, 0.98, and 0.99). The size distributions are between the radius limits $r_{min} = 0.005\,\mu\text{m}$ and $r_{max} = 10.0\,\mu\text{m}$. Note that mineral and water-insoluble aerosols have no relative humidity induced swelling.

The aerosol components can be combined to form aerosol models, where for each of them, the number of components $N$ and the corresponding number mixing ratios $w_i$ have to be specified. In the present application, we use the 10 aerosol models proposed in the OPAC database [6].

### 3.1.2. Set 2

The second model set includes the following aerosol components:

1. *black carbon,*
2. *dust,*
3. *organic carbon,*
4. *sea salt,* and
5. *sulfate.*

As in the previous case, we generate a database that gives the values of $\overline{C}_{ext}$, $\overline{C}_{sct}$, $\overline{\omega}$, and $\overline{a}_n$, for (i) each aerosol component, (ii) a set of wavelengths (61 values in the range $0.250\,\mu\text{m}$–$40\,\mu\text{m}$), and (iii) a set of relative humidities (36 values: 0.00, 0.05, ..., 0.80, 0.81, 0.82, ..., 0.99). The aerosol single scattering properties are calculated based on the formalism presented in Ref. [14]. The following peculiarities of the computational process can be pointed out.

1. A log-normal distribution with radius between $0.005\,\mu\text{m}$ and $0.3\,\mu\text{m}$ is assumed for black carbon, organic carbon and sulfate, with size parameters as in Table 2 of Ref. [14].
2. The optical properties of dust and sea salt are computed across each of the five size bins used in the simulation. For dust, a constant number size distribution across the bin is assumed, while for sea salt, a functional form of the particle size distribution taken from Ref. [15] is considered. The dry size bins for dust are between the following radius limits (in $\mu\text{m}$): 0.1–1, 1–1.6, 1.6–3, 3–6, 6–10. For sea salt, the limits are (in $\mu\text{m}$): 0.03–0.1, 0.1–0.5, 0.5–1.5, 1.5–5, 5–10.
3. Dust has no relative humidity induced swelling. Black carbon, organic carbon, sea salt and sulfate have relative humidity dependent growth factors (ratio of wet to dry particle diameter). The growth factors for organic carbon and sulfate are taken from the OPAC database [6], with modifications as in Ref. [14]. The empirical relationship of Gerber [16] is used for obtaining growth factors for sea salt.
4. The spectral complex refractive index is taken from the OPAC database. The scattering characteristics are computed using the Mie theory, except for dust, where a precomputed database for ellipsoidal particles is considered [17].

For this application, we use the 10 aerosol models (mixtures of sulfate, dust, sea salt, black carbon, and organic carbon) obtained by a cluster analysis using the Goddard Chemistry Aerosol Radiation

and Transport (GOCART) model [7]. This uses data from global circulation or chemical transport models employing the sensitivity of multi-angle imaging to natural mixtures of aerosols. Multi-angle imaging is sensitive to aerosol optical depth and aerosol type and this principle is employed here.

### 3.1.3. Set 3

The third set comprises the aerosol models included in the OMI multiwavelength aerosol retrieval algorithm [8]. There are five major aerosol types, where each type consists of several aerosol models depending on their optical properties and particle size distribution. On a global scale, four main tropospheric aerosol types can be distinguished: (i) urban-industrial aerosols originating from fossil fuel combustion, (ii) carbonaceous aerosols generated from natural and anthropogenic biomass burning, (iii) desert dust aerosols, injected into the atmosphere by winds, and (iv) naturally produced oceanic aerosols. After major volcanic eruptions, the aerosol optical thickness of the stratosphere can be significantly increased for several years. For this reason, a volcanic aerosol type is also included.

### 3.1.4. Set 4

The fourth set comprises the aerosol models included in the MODIS aerosol retrieval algorithm [3]. These are derived by performing a cluster analysis on the entire time series of almucantur aerosol properties from global AERONET sites. There are three fine-dominated (spherical) and one coarse-dominated (spheroid) aerosol optical models that represent the range of likely and observable global aerosol conditions. The fine-dominated aerosol models differ mainly in their values of single scattering albedo $\overline{\omega}_m$, i.e., moderately absorbing ($\overline{\omega}_m = 0.90$), absorbing ($\overline{\omega}_m = 0.85$), and non-absorbing ($\overline{\omega}_m = 0.95$). Each aerosol model is bilognormal, with dynamic (function of optical depth) size parameters (radius, standard deviation, volume distribution) and complex refractive index.

### 3.1.5. Set 5

The fifth set includes the aerosol models derived from several AERONET sites [5]. At the selected aerosol sites, with well-known meteorological and environmental conditions, the following aerosol models are identified: (i) urban-industrial from fossil fuel combustion in populated industrial regions, (ii) biomass burning produced by forest and grassland fires, (iii) desert dust blown into the atmosphere by wind, and (iv) aerosol of marine origin. In addition, a mixed aerosol model over the Maldives is included. The parameters of each aerosol model have dynamic variability as a function of optical thickness.

## 4. Numerical Simulations

In this section, we apply the algorithm to the retrieval of aerosol optical thickness and layer height by generating synthetic measurements corresponding to the EPIC instrument. The retrieval is performed under the assumption that the surface albedo is known.

The state vector $\mathbf{x}$ comprises the aerosol optical thickness $\tau$ and layer height $H$, i.e., $\mathbf{x} = [\tau, H]^T$. As in Ref. [1], the true aerosol optical thicknesses to be retrieved are $\tau_t = 0.25, 0.50, 0.75, 1.0, 1.25$, and 1.5, while for the true aerosol layer height, we take $H_t = 1.0, 1.5, 2.0, 2.5$, and 3.0 km. The a priori values, which coincide with the initial guesses, are $\tau_a = 2.0$ and $H_a = 4$ km, while the surface albedo is $A = 0.06$. The solution accuracy is characterized by the relative errors

$$\varepsilon^\tau_{\text{mean}} = \frac{|\tau_{\text{mean}} - \tau_t|}{\tau_t} \text{ and } \varepsilon^H_{\text{mean}} = \frac{|H_{\text{mean}} - H_t|}{\tau_t}$$

corresponding to the mean solution estimate (cf. Equation (3)) $\widehat{\mathbf{x}}^\delta_{\text{mean}} = [\tau_{\text{mean}}, H_{\text{mean}}]$, and

$$\varepsilon^\tau_{\text{max}} = \frac{|\tau_{\text{max}} - \tau_t|}{\tau_t} \text{ and } \varepsilon^H_{\text{max}} = \frac{|H_{\text{max}} - H_t|}{\tau_t}$$

those corresponding to the maximum solution estimate (cf. Equation (4)) $\widehat{\mathbf{x}}_{max}^{\delta} = [\tau_{max}, H_{max}]$. For the mean solution estimate, the average relative errors over $\tau_t$ for $H_t = 3.0$ km are defined by

$$\varepsilon_{mean}^{\tau(\tau)} = \frac{1}{N_\tau} \sum_{i=1}^{N_\tau} \varepsilon_{mean i}^\tau \text{ and } \varepsilon_{mean}^{H(\tau)} = \frac{1}{N_\tau} \sum_{i=1}^{N_\tau} \varepsilon_{mean i}^H,$$

where $N_\tau = 6$, while the average relative errors over $H_t$ for $\tau_t = 1.0$ are defined by

$$\varepsilon_{mean}^{\tau(H)} = \frac{1}{N_H} \sum_{i=1}^{N_H} \varepsilon_{mean i}^\tau \text{ and } \varepsilon_{mean}^{H(H)} = \frac{1}{N_H} \sum_{i=1}^{N_H} \varepsilon_{mean i}^H,$$

where $N_H = 5$. For the maximum solution estimate, similar relations are valid for $\varepsilon_{max}^{\tau(\tau)}, \varepsilon_{max}^{H(\tau)}, \varepsilon_{max}^{\tau(H)}$, and $\varepsilon_{nax}^{H(H)}$.

To analyze the accuracy of the aerosol retrieval, we consider three test examples. In the first example, the synthetic measurements are generated by choosing as truth, the urban industrial model derived from AERONET measurements at the Goddard Space Flight Center (GSFC) in Greenbelt, Maryland; in the second example, the true model is the mixed aerosol model derived from AERONET measurements over the Maldives; in the third example, the true model is the biomass burning model derived from AERONET measurements over the African savanna in Zambia. For each test, we consider extended and reduced sets of aerosol models. These are illustrated in Table 1. The reduced sets of aerosol models require less computational time and, in principle, should be sufficient to reproduce the true model. Since the OPAC and GOCART aerosol models depend on the relative humidity $U$, these models are applied with $U = 0.80, 0.90$, and $0.95$.

**Table 1.** Extended (upper table) and reduced (lower table) sets of aerosol models. The reduced OMI set of aerosol models labeled (1) is used for the first and second test examples, while the reduced set labeled (2) is used for the third example.

| OPAC | GOCART | OMI | MODIS |
|---|---|---|---|
| Cont. clean Cont. average Cont. pol. Urban Desert | Sulfurous dusty smoke Dusty sulfate Dust Sulfurous smoke Sulfurous dust Marine dusty sulfate Sulfate Smokey sulfate | WA120X WA130X BB210X BB220X | Nonabs. Modabs. Abs. Dust |
| **OPAC** | **GOCART** | **OMI** | **MODIS** |
| Cont. clean Cont. average Cont. pol. Urban | Sulfurous dusty smoke Dusty sulfate Sulfurous smoke Sulfurous dust Sulfate Smokey sulfate | $\begin{pmatrix} \text{WA120X} \\ \text{WA130X} \end{pmatrix}^{(1)}$ $\begin{pmatrix} \text{BB210X} \\ \text{BB220X} \end{pmatrix}^{(2)}$ | Nonabs. Modabs. Abs. |

The relative errors for the three test examples corresponding to the extended and reduced sets of aerosol models are given in Figures 1–4.

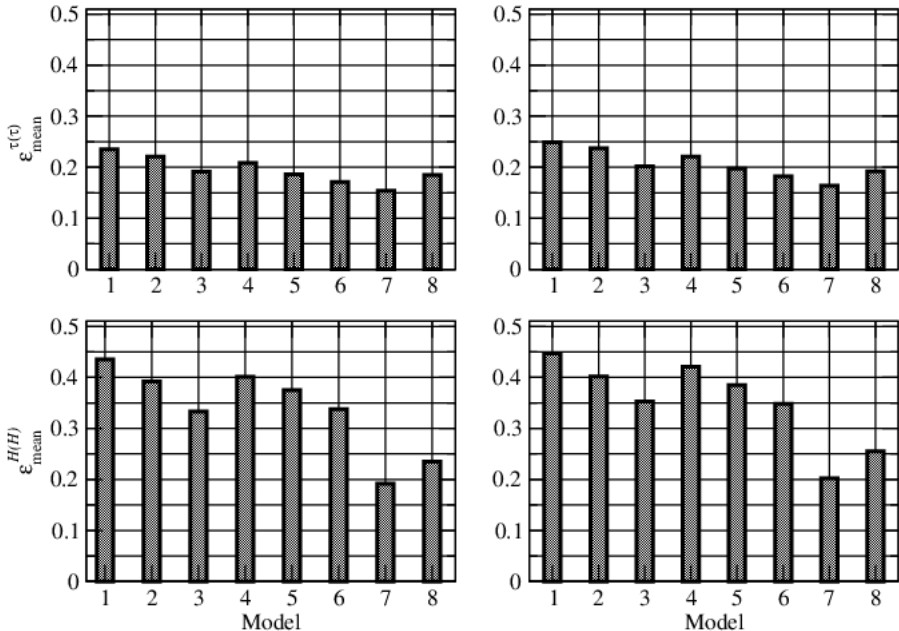

**Figure 1.** Relative errors $\varepsilon_{\text{mean}}^{\tau(\tau)}$ and $\varepsilon_{\text{mean}}^{H(H)}$ for the first test example. The aerosol databases are OPAC with $U = 0.80$ (1), $U = 0.90$ (2), and $U = 0.95$ (3), GOCART with $U = 0.80$ (4), $U = 0.90$ (5), and $U = 0.95$ (6), OMI (7), and MODIS (8). The plots in the left panels correspond to the extended sets of aerosol models, while those in the right panels correspond to the reduced sets.

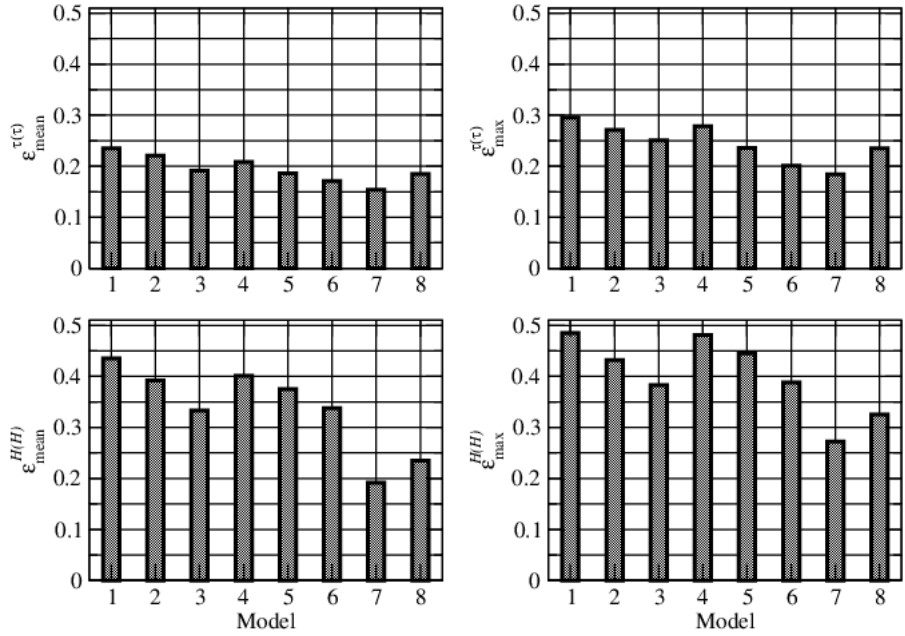

**Figure 2.** Relative errors $\varepsilon_{\text{mean}}^{\tau(\tau)}$, $\varepsilon_{\text{max}}^{\tau(\tau)}$, $\varepsilon_{\text{mean}}^{H(H)}$, and $\varepsilon_{\text{max}}^{H(H)}$ for the first test example using the extended sets of aerosol models. The aerosol databases are labeled in the same way as in Figure 1.

The following inferences can be drawn.

1.  For the first test example, the best results correspond to the OMI aerosol models (the relative errors are $\varepsilon_{\text{mean}}^{\tau(\tau)} = 0.154$ and $\varepsilon_{\text{mean}}^{H(H)} = 0.192$). For the second and third test examples, the best results correspond to the MODIS aerosol models (the relative errors are $\varepsilon_{\text{mean}}^{\tau(\tau)} = 0.202$ and $\varepsilon_{\text{mean}}^{H(H)} = 0.224$ for the second case, and $\varepsilon_{\text{mean}}^{\tau(\tau)} = 0.168$ and $\varepsilon_{\text{mean}}^{H(H)} = 0.205$ for the third case), followed by the OMI aerosol models. The latter result is not surprising because the MODIS aerosol models are the result of a cluster analysis of aerosol retrievals from global AERONET sites.

Note that aerosol optical thickness errors are smaller, but not significantly smaller, than errors in the layer height.

2. For the OPAC and GOCART aerosol models, the best fits correspond to a high and unrealistic value of relative humidity, $U = 0.95$. Note that the annual relative humidity in Greenbelt, Maryland is about 0.64; the corresponding value for Male, Maldives is about 0.79, and that for Lusaka, Zambia is about 0.61 (with a maximum of 0.80 in January).

3. The results for the extended set of aerosol models are slightly better than those for the reduced set.

4. From Figure 2, it is apparent that $\varepsilon_{\mathrm{max}}^{\tau(\tau)} > \varepsilon_{\mathrm{mean}}^{\tau(\tau)}$ and $\varepsilon_{\mathrm{max}}^{H(H)} > \varepsilon_{\mathrm{mean}}^{H(H)}$; thus, as expected, the maximum solution estimate $\widehat{\mathbf{x}}_{\mathrm{max}}^{\delta} = [\tau_{\mathrm{max}}, H_{\mathrm{max}}]$ is less accurate than the mean solution estimate $\widehat{\mathbf{x}}_{\mathrm{mean}}^{\delta} = [\tau_{\mathrm{mean}}, H_{\mathrm{mean}}]$.

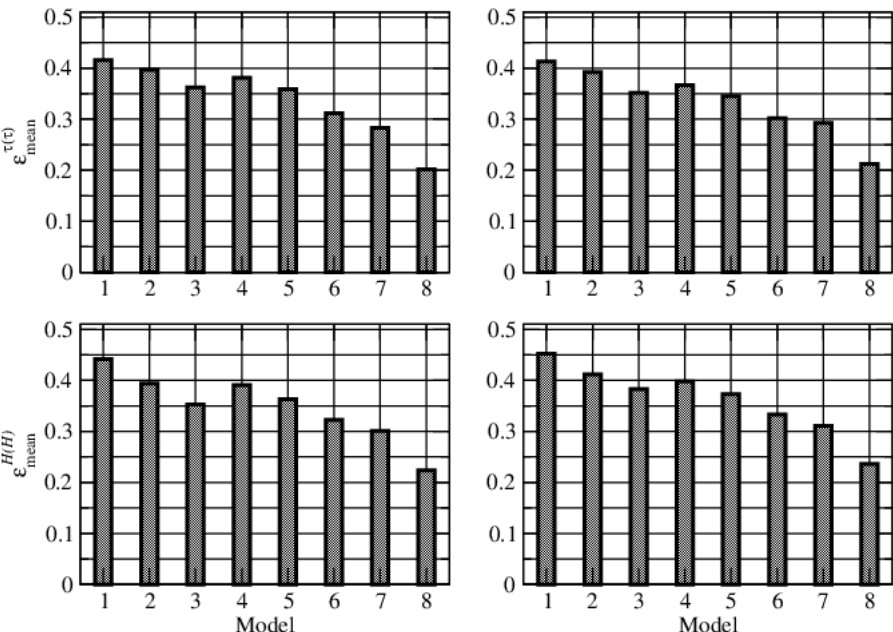

**Figure 3.** Same as Figure 1 but for the second test example.

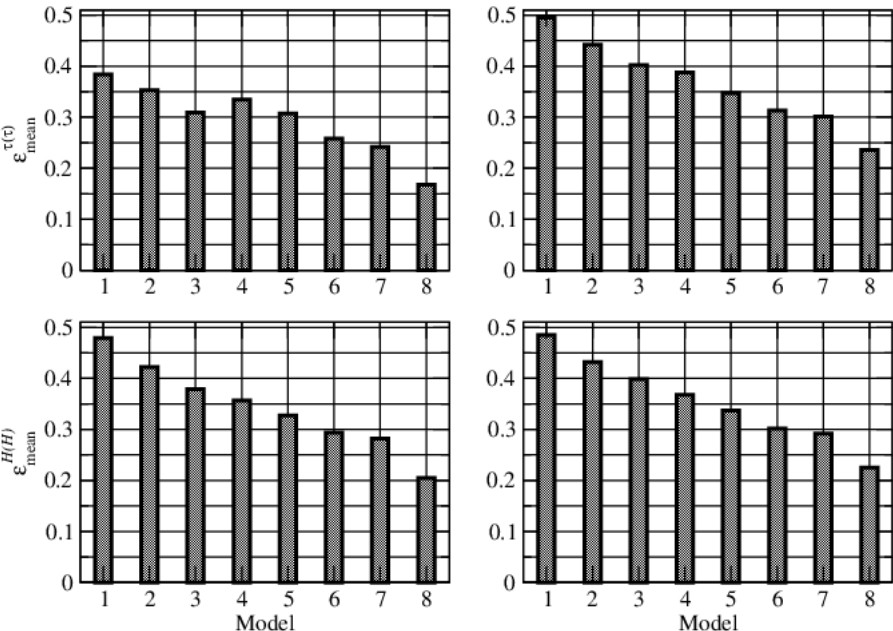

**Figure 4.** Same as in Figure 1 but for the third test example.

In the retrieval, the input parameters of the forward (radiative transfer) model are the single scattering albedo and the expansion coefficients of the phase function. Therefore, in order to explain the above findings, we show in Table 2 the single scattering albedo $\overline{\omega}_m$, the asymmetry parameter $\overline{g}_m$, and the relative error $\varepsilon^{\tau}_{\text{mean}}$ for a particular retrieval case of the first test example ($\tau_t = 1.0$ and $H_t = 3.0$ km). The corresponding phase functions are illustrated in Figure 5. Taking into account the representation of the single scattering radiance, we deduce that the backscattering region of the phase function (the angular domain around the viewing direction) is relevant for our analysis. The results can be summarized as follows.

1.  The best retrievals correspond to the OMI, followed by the GOCART-0.95, aerosol models.
2.  The single scattering albedo $\overline{\omega}_m$ for the AERONET aerosol model is best approximated by the scattering albedos associated with the GOCART-0.80 and OPAC-0.90 aerosol models. However, all values of $\overline{\omega}_m$ are above 0.95, so that roughly speaking, all aerosol models are nonabsorbing. Therefore, we believe that, for the present application, $\overline{\omega}_m$ is not a relevant indicator of the goodness of fit.
3.  The asymmetry parameter $\overline{g}_m$ for the AERONET aerosol model is best approximated by the asymmetry parameters associated with the OMI and GOCART-0.95 aerosol models. Moreover, the phase function in the backscattering region is also accurately reproduced by the same models. Thus, the accuracy of the asymmetry parameter, which is the first-order moment of the phase function, and the phase function in a region around the viewing direction, determine the retrieval errors.
4.  From Figure 5, it is also apparent that (i) GOCART-0.95 is superior to GOCART-0.80 and GOCART-0.90, and (ii) OPAC-0.95 is superior to OPAC-0.80 and OPAC-0.90. Thus, for these aerosol databases, better approximations for the asymmetry parameter and the phase function in the backscattering region correspond to larger values of $U$. Unfortunately, as previously mentioned, these large values of $U$ are physically unrealistic. The conclusion which can be drawn is that climatologically correct GOCART and OPAC aerosol models are not always the best option for retrievals.

**Table 2.** Single scattering albedo $\overline{\omega}_m$, asymmetry parameter $\overline{g}_m$, and relative error $\varepsilon^{\tau}_{\text{mean}}$ for $\tau_t = 1.0$ and $H_t = 3.0$ km. The results correspond to the first test example and the extended sets of aerosol models.

| Aerosol Models | $\overline{\omega}_{\mathbf{m}}$ | $\overline{g}_{\mathbf{m}}$ | $\varepsilon^{\tau}_{\mathbf{mean}}$ |
|---|---|---|---|
| AERONET | $9.765 \times 10^{-1}$ | $7.327 \times 10^{-1}$ | - |
| OPAC-0.80 | $9.618 \times 10^{-1}$ | $6.572 \times 10^{-1}$ | 0.235 |
| OPAC-0.90 | $9.743 \times 10^{-1}$ | $6.776 \times 10^{-1}$ | 0.221 |
| OPAC-0.95 | $9.836 \times 10^{-1}$ | $6.961 \times 10^{-1}$ | 0.185 |
| GOCART-0.80 | $9.753 \times 10^{-1}$ | $6.906 \times 10^{-1}$ | 0.201 |
| GOCART-0.90 | $9.826 \times 10^{-1}$ | $6.994 \times 10^{-1}$ | 0.186 |
| GOCART-0.95 | $9.871 \times 10^{-1}$ | $7.139 \times 10^{-1}$ | 0.171 |
| OMI | $9.672 \times 10^{-1}$ | $7.321 \times 10^{-1}$ | 0.152 |
| MODIS | $9.674 \times 10^{-1}$ | $6.789 \times 10^{-1}$ | 0.191 |

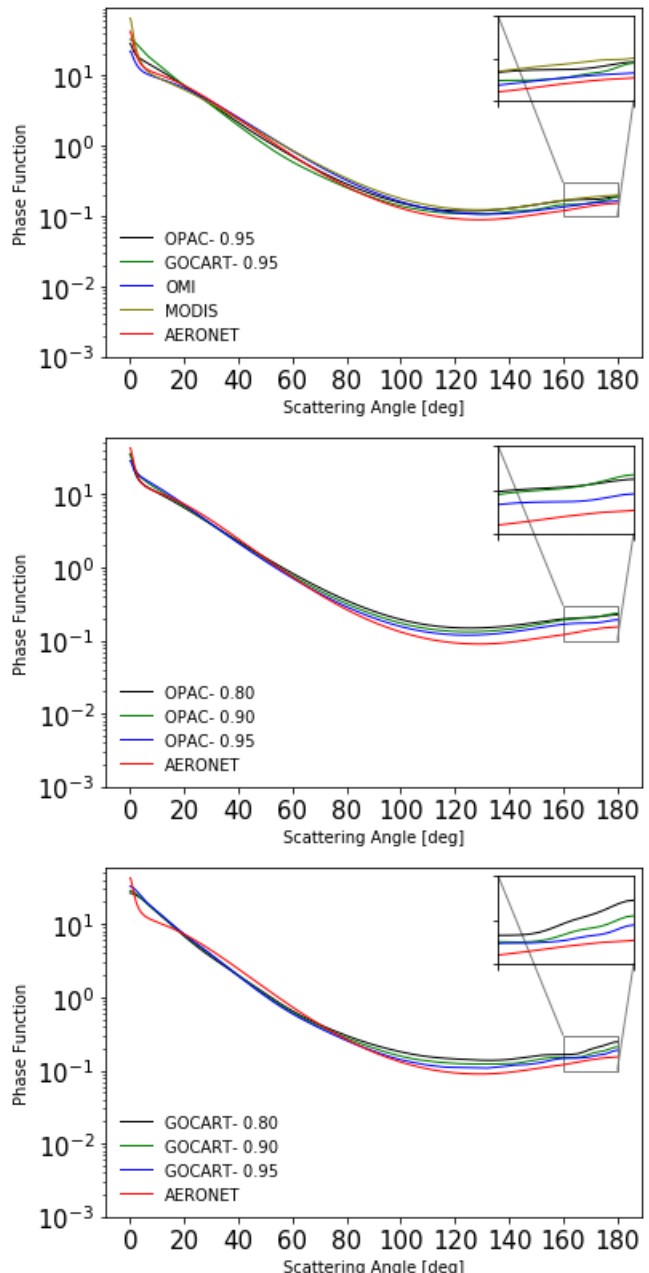

**Figure 5.** Phase functions for $\tau_t = 1.0$ and $H_t = 3.0$. The results correspond to the first test example and the extended sets of aerosol models.

## 5. Conclusions

This paper presents the results of aerosol retrieval using an algorithm that takes into account the uncertainty in aerosol model selection. The solution corresponding to a specific aerosol model is characterized by the relative evidence, which is a measure of how well the aerosol model fits the measurement. Based on this quantity, the following solution estimates are used: the maximum solution estimate, corresponding to the aerosol model with the highest evidence, and the mean solution estimate, representing a linear combination of solutions weighted by their evidences.

We generated synthetic measurements for the EPIC instrument using as reference certain aerosol models derived from AERONET measurements at different sites. In the retrieval, the surface albedo was assumed to be known, and four aerosol models, included in the OPAC, GOCART, OMI, and MODIS databases, were evaluated. The goal of our numerical analysis was to quantify errors in the retrieval of

aerosol optical thickness and layer height, when the true aerosol model is unknown. The following conclusions were drawn.

1.  For the first test example, the best results correspond to the OMI aerosol models, while for the second and third test examples, the best results correspond to the MODIS, followed by the OMI, aerosol models.
2.  The maximum solution estimate is less accurate than the mean solution estimate.
3.  On average, the best relative retrieval errors, corresponding to the mean solution estimate, are about 0.20 for both the aerosol optical thickness and layer height.
4.  The best aerosol model is the one with (i) a value of the asymmetry parameter and (ii) an angular variation of the phase function around the viewing direction that are close to the values corresponding to the reference aerosol model. These criteria can also be fulfilled by aerosol models that are climatologically incorrect.

Since the MODIS aerosol models are derived from AERONET measurements, it seems that in practice, the OMI aerosol models are the most suitable option. However, the retrieval errors in the mean solution estimate are large (on average, larger than 0.2). These can be improved if, instead of a single-angle measurement, a multi-angle measurement approach is used. In this case, the weighting factors for each aerosol component are included in the retrieval.

**Author Contributions:** Conceptualization, V.N. and A.D.; Data curation, S.S.; Formal analysis, S.S. and V.M.G.; Funding acquisition, V.N., D.L. and A.D.; Investigation, S.S.; Methodology, V.N. and A.D.; Project administration, D.L. and A.D.; Supervision, V.N., Dmitry S. Efremenko and A.D.; Validation, V.N. and A.D.; Writing—original draft, S.S.; Writing—review & editing, S.S., V.N., D.S.E., D.L. and A.D. All authors have read and agreed to the published version of the manuscript.

**Funding:** This research was funded by the German Aerospace Center (DLR) and the German Academic Exchange Service (DAAD) through the programme DLR/DAAD Research Fellowships 2015 (57186656), with reference numbers 91613528 and 91627488.

**Acknowledgments:** A portion of this research was carried out at the Jet Propulsion Laboratory, California Institute of Technology, under a contract with the National Aeronautics and Space Administration (80NM0018D0004). V. N. acknowledges support from the NASA Earth Science US Participating Investigator program (solicitation NNH16ZDA001N-ESUSPI).

**Conflicts of Interest:** There are no conflicts of interest.

## Appendix A

In this appendix, we summarize the aerosol model sets used in this study.

**Table A1.** Geometrical and optical properties of the aerosol models included in the OPAC database. The relative refractive index corresponds to the wavelength $\lambda = 750\,\text{nm}$ and the relative humidity $U = 0.8$.

| Model | Component | $r_{\text{mod}}\,(\mu\text{m})$ | $s = e^{\sigma}$ | $m = (\text{Re(m)}, \text{Im(m)})$ | $w_i$ |
|---|---|---|---|---|---|
| Cont. clean | Water sol. | 0.0212 | 2.24 | $(1.40, 2.83 \times 10^{-3})$ | 1.0 |
| | Water insol. | 0.4710 | 2.51 | $(1.53, 8.0 \times 10^{-3})$ | $0.577 \times 10^{-4}$ |
| Cont. avrg. | Water sol. | 0.0212 | 2.24 | $(1.40, 2.83 \times 10^{-3})$ | 0.95 |
| | Soot | 0.0118 | 2.00 | $(1.75, 4.3 \times 10^{-1})$ | 0.05 |
| | Water insol. | 0.4710 | 2.51 | $(1.53, 8.0 \times 10^{-3})$ | $0.261 \times 10^{-4}$ |
| Cont. pol. | Water sol. | 0.0212 | 2.24 | $(1.40, 2.83 \times 10^{-3})$ | 0.90 |
| | Soot | 0.0118 | 2.00 | $(1.75, 4.3 \times 10^{-1})$ | 0.10 |
| | Water insol. | 0.4710 | 2.51 | $(1.53, 8.0 \times 10^{-3})$ | $0.12 \times 10^{-4}$ |
| Urban | Water sol. | 0.0212 | 2.24 | $(1.40, 2.83 \times 10^{-3})$ | 0.80 |
| | Soot | 0.0118 | 2.00 | $(1.75, 4.3 \times 10^{-1})$ | 0.20 |
| | Water insol. | 0.4710 | 2.51 | $(1.53, 8.0 \times 10^{-3})$ | $0.949 \times 10^{-5}$ |

**Table A1.** *Cont.*

| Model | Component | $r_{\mathbf{mod}}\,(\mathbf{\mu m})$ | $s = e^{\sigma}$ | $m = (\mathbf{Re(m), Im(m)})$ | $w_i$ |
|---|---|---|---|---|---|
| Desert | Water sol. | 0.0212 | 2.24 | $(1.40, 2.83 \times 10^{-3})$ | 0.87 |
| | Mineral nuc. | 0.0700 | 1.95 | $(1.53, 4.00 \times 10^{-3})$ | 0.117 |
| | Mineral acc. | 0.3900 | 2.00 | $(1.53, 4.00 \times 10^{-3})$ | $0.133 \times 10^{-1}$ |
| | Mineral coa. | 1.9000 | 2.15 | $(1.53, 4.00 \times 10^{-3})$ | $0.617 \times 10^{-4}$ |
| Marit. clean | Water sol. | 0.0212 | 2.24 | $(1.40, 2.83 \times 10^{-3})$ | 0.987 |
| | See salt acc. | 0.2090 | 2.03 | $(1.35, 2.73 \times 10^{-7})$ | $0.132 \times 10^{-1}$ |
| | See salt coa. | 1.7500 | 2.03 | $(1.35, 2.72 \times 10^{-7})$ | $0.211 \times 10^{-5}$ |
| Marit. pol. | Water sol. | 0.0212 | 2.24 | $(1.40, 2.83 \times 10^{-3})$ | 0.422 |
| | Soot | 0.0118 | 2.00 | $(1.75, 4.3 \times 10^{-1})$ | 0.576 |
| | See salt acc. | 0.2090 | 2.03 | $(1.35, 2.73 \times 10^{-7})$ | $0.222 \times 10^{-2}$ |
| | See salt coa. | 1.7500 | 2.03 | $(1.35, 2.72 \times 10^{-7})$ | $0.356 \times 10^{-6}$ |
| Marit. trop. | Water sol. | 0.0212 | 2.24 | $(1.40, 2.83 \times 10^{-3})$ | 0.983 |
| | See salt acc. | 0.2090 | 2.03 | $(1.35, 2.73 \times 10^{-7})$ | $0.167 \times 10^{-1}$ |
| | See salt coa. | 1.7500 | 2.03 | $(1.35, 2.72 \times 10^{-7})$ | $0.217 \times 10^{-5}$ |
| Arctic | Water sol. | 0.0212 | 2.24 | $(1.40, 2.83 \times 10^{-3})$ | 0.197 |
| | Soot | 0.0118 | 2.00 | $(1.75, 4.3 \times 10^{-1})$ | 0.803 |
| | Water insol. | 0.4710 | 2.51 | $(1.53, 8.0 \times 10^{-3})$ | $0.152 \times 10^{-5}$ |
| | See salt acc. | 0.2090 | 2.03 | $(1.35, 2.73 \times 10^{-7})$ | $0.288 \times 10^{-3}$ |
| Antarctic | Sulfate | 0.0695 | 2.03 | $(1.35, 1.39 \times 10^{-7})$ | 0.998 |
| | See salt acc. | 0.2090 | 2.03 | $(1.35, 2.73 \times 10^{-7})$ | $0.109 \times 10^{-2}$ |
| | Mineral tra. | 0.5000 | 2.20 | $(1.530, 4.0 \times 10^{-3})$ | $0.123 \times 10^{-3}$ |

**Table A2.** GOCART aerosol models and the weighting factors (in percentage) corresponding to the aerosol components: sulfate (SS), dust (DU), see salt (SS), black carbon (BC), and organic carbon (OC).

| Model | Component | | | | |
|---|---|---|---|---|---|
| | SU | DU | SS | BC | OC |
| Sulfurous dusty smoke | 27.4 | 30.7 | 5.9 | 5.9 | 30.1 |
| Marine sulfate | 44.6 | 4.7 | 36.7 | 3.0 | 11.0 |
| Dusty sulfate | 54.7 | 25.6 | 7.2 | 3.4 | 9.1 |
| Dust | 13.0 | 80.2 | 1.1 | 1.7 | 4.0 |
| Sulfurous smoke | 29.7 | 6.0 | 3.1 | 9.3 | 51.8 |
| Sulfurous dust | 31.0 | 53.1 | 4.7 | 3.2 | 8.0 |
| Marine dusty sulfate | 43.1 | 27.0 | 19.5 | 3.2 | 7.2 |
| Sulfate | 66.1 | 4.7 | 14.1 | 3.4 | 11.6 |
| Sulfurous marine | 28.8 | 3.8 | 58.4 | 1.7 | 7.3 |
| Smokey sulfate | 45.0 | 6.8 | 14.0 | 6.7 | 27.5 |

**Table A3.** Geometrical and optical properties of the aerosol models considered in the OMI multiwavelength aerosol retrieval algorithm.

| Type | Model | $r_{\mathbf{mod}}(\mu\mathbf{m})$ | $s = \mathbf{e}^{\sigma}$ | $\mathbf{m} = (\mathbf{Re(m)}, \mathbf{Im(m)})$ | $w_{\mathbf{coarse}}$ |
|---|---|---|---|---|---|
| Weakly absorbing | WA1101 | 0.078 0.497 | 1.499 2.160 | $(1.4, 5.0 \times 10^{-8})$ | $4.36 \times 10^{-4}$ |
| | WA1102 | 0.088 0.509 | 1.499 2.160 | $(1.4, 5.0 \times 10^{-8})$ | $4.04 \times 10^{-4}$ |
| | WA1103 | 0.137 0.567 | 1.499 2.160 | $(1.4, 5.0 \times 10^{-8})$ | $8.10 \times 10^{-4}$ |
| | WA1104 | 0.030 0.240 | 2.030 2.030 | $(1.4, 5.0 \times 10^{-8})$ | $1.53 \times 10^{-2}$ |
| | WA1201 | 0.078 0.497 | 1.499 2.160 | $(1.4, 4.0 \times 10^{-3})$ | $4.36 \times 10^{-4}$ |
| | WA1202 | 0.088 0.509 | 1.499 2.160 | $(1.4, 4.0 \times 10^{-3})$ | $4.04 \times 10^{-4}$ |
| | WA1203 | 0.126 0.421 | 1.499 2.160 | $(1.4, 4.0 \times 10^{-3})$ | $8.10 \times 10^{-4}$ |
| | WA1301 | 0.078 0.497 | 1.499 2.160 | $(1.4, 1.2 \times 10^{-2})$ | $4.36 \times 10^{-4}$ |
| | WA1302 | 0.088 0.509 | 1.499 2.160 | $(1.4, 1.2 \times 10^{-2})$ | $4.04 \times 10^{-4}$ |
| | WA1303 | 0.137 0.567 | 1.499 2.160 | $(1.4, 1.2 \times 10^{-2})$ | $8.10 \times 10^{-4}$ |
| Biomass burning | BB2101 | 0.074 0.511 | 1.537 2.203 | $(1.5, 1.0 \times 10^{-2})$ | $1.70 \times 10^{-4}$ |
| | BB2102 | 0.087 0.567 | 1.537 2.203 | $(1.5, 1.0 \times 10^{-2})$ | $2.06 \times 10^{-4}$ |
| | BB2103 | 0.124 0.719 | 1.537 2.203 | $(1.5, 1.0 \times 10^{-2})$ | $2.94 \times 10^{-4}$ |
| | BB2201 | 0.074 0.511 | 1.537 2.203 | $(1.5, 2.0 \times 10^{-2})$ | $1.70 \times 10^{-4}$ |
| | BB2202 | 0.087 0.567 | 1.537 2.203 | $(1.5, 2.0 \times 10^{-2})$ | $2.06 \times 10^{-4}$ |
| | BB2203 | 0.124 0.719 | 1.537 2.203 | $(1.5, 2.0 \times 10^{-2})$ | $2.94 \times 10^{-4}$ |
| | BB2301 | 0.074 0.511 | 1.537 2.203 | $(1.5, 3.0 \times 10^{-2})$ | $1.70 \times 10^{-4}$ |
| | BB2302 | 0.087 0.567 | 1.537 2.203 | $(1.5, 3.0 \times 10^{-2})$ | $2.06 \times 10^{-4}$ |
| | BB2303 | 0.124 0.719 | 1.537 2.203 | $(1.5, 3.0 \times 10^{-2})$ | $2.94 \times 10^{-4}$ |
| Desert dust | DD3101 | 0.042 0.670 | 1.697 1.806 | $(1.53, 4.0 \times 10^{-3})$ | $4.35 \times 10^{-3}$ |
| | DD3102 | 0.052 0.670 | 1.697 1.806 | $(1.53, 4.0 \times 10^{-3})$ | $4.35 \times 10^{-3}$ |
| | DD3201 | 0.042 0.670 | 1.697 1.806 | $(1.53, 1.0 \times 10^{-2})$ | $4.35 \times 10^{-3}$ |
| | DD3202 | 0.052 0.670 | 1.697 1.806 | $(1.53, 1.0 \times 10^{-2})$ | $4.35 \times 10^{-3}$ |

**Table A3.** *Cont.*

| Type | Model | $r_{\mathbf{mod}}(\mu m)$ | $s = e^\sigma$ | $m = (Re(m), Im(m))$ | $w_{\mathbf{coarse}}$ |
|---|---|---|---|---|---|
| Maritime | Maritime mod. abs. | 0.030 0.240 | 2.030 2.030 | $(1.4, 4.0 \times 10^{-3})$ $(1.4, 5.0 \times 10^{-8})$ | $1.55 \times 10^{-4}$ |
| | Maritime abs. | 0.030 0.240 | 2.030 2.030 | $(1.4, 1.2 \times 10^{-2})$ $(1.4, 5.0 \times 10^{-8})$ | $1.55 \times 10^{-4}$ |
| | Maritime clean | 0.030 0.240 | 2.030 2.030 | $(1.4, 5.0 \times 10^{-8})$ | $1.53 \times 10^{-2}$ |
| Volcanic | VO4101 | 0.230 0.230 | 0.800 0.800 | $(1.45, 7.5 \times 10^{-7})$ | 0.5 |

**Table A4.** Geometrical and optical properties of the aerosol models considered in the MODIS aerosol retrieval algorithm. The four values of the refractive index for dust correspond to the wavelengths $\lambda = 0.470, 0.550, 0.660, 2.100\,\mu m$.

| Model | $r_{\mathbf{v}}(\mu m)$ | $\sigma$ | $m = (Re(m), Im(m))$ | $V_0(\mu m^3/\mu m^2)$ |
|---|---|---|---|---|
| Nonabs. | $0.160 + 0.0434\tau$ $3.325 + 0.1411\tau$ | $0.364 + 0.1529\tau$ $0.759 + 0.0168\tau$ | $(1.42, 0.004\text{–}0.0015\tau)$ | $0.1718\tau^{0.821}$ $0.0934\tau^{0.639}$ |
| Modabs. | $0.145 + 0.0203\tau$ $3.101 + 0.3364\tau$ | $0.374 + 0.1365\tau$ $0.729 + 0.098\tau$ | $(1.43, 0.008\text{–}0.002\tau)$ | $0.1642\tau^{0.775}$ $0.1482\tau^{0.684}$ |
| Abs. | $0.134 + 0.0096\tau$ $3.448 + 0.9489\tau$ | $0.383 + 0.0794\tau$ $0.743 + 0.0409\tau$ | $(1.51, 0.02)$ | $0.1748\tau^{0.891}$ $0.1043\tau^{0.682}$ |
| Dust | $0.1416\tau^{-0.052}$ $2.2$ | $0.7561\tau^{0.148}$ $0.554\tau^{-0.052}$ | $(1.48\tau^{-0.021}, 0.0025\tau^{0.132})$ $(1.48\tau^{-0.021}, 0.002)$ $(1.48\tau^{-0.021}, 0.0018\tau^{-0.08})$ $(1.46\tau^{-0.021}, 0.0018\tau^{-0.30})$ | $0.0871\tau^{1.026}$ $0.6786\tau^{1.057}$ |

**Table A5.** Geometrical and optical properties of the aerosol models derived at several AERONET sites. Here, $\alpha$ is the Angström coefficient and the four values of the refractive index for desert dust correspond to the wavelengths $\lambda = 0.470, 0.550, 0.660, 2.100\,\mu m$.

| Model | $r_{\mathbf{v}}(\mu m)$ | $\sigma$ | $m = (Re(m), Im(m))$ | $V_0(\mu m^3/\mu m^2)$ | $\alpha$ |
|---|---|---|---|---|---|
| Urban industrial (GSFC, Greenbelt, 1993–2000) | $0.12 + 0.11\tau_{440}$ $3.03 + 0.49\tau_{440}$ | 0.38 0.75 | $(1.41\text{–}0.03\tau_{440}, 0.003)$ | $0.15\tau_{440}$ $0.01 + 0.05\tau_{440}$ | 1.90 |
| Urban industrial (Mexico City, 1999–2000) | $0.12 + 0.04\tau_{440}$ $2.72 + 0.60\tau_{440}$ | 0.43 0.63 | $(1.47, 0.014)$ | $0.12\tau_{440}$ $0.11\tau_{440}$ | 1.80 |
| Mixed (Maldives, 1999–2000) | $0.18$ $2.62 + 0.61\tau_{440}$ | 0.46 0.76 | $(1.44, 0.011)$ | $0.12\tau_{440}$ $0.15\tau_{440}$ | 1.55 |
| Biomass burning (African savanna, Zambia, 1995–2000) | $0.12 + 0.025\tau_{440}$ $3.22 + 0.710\tau_{440}$ | 0.40 0.73 | $(1.51, 0.021)$ | $0.12\tau_{440}$ $0.09\tau_{440}$ | 1.95 |
| Biomass burning (Boreal forest, USA and Canada, 1994–1998) | $0.15 + 0.015\tau_{440}$ $3.22 + 0.710\tau_{440}$ | 0.43 0.81 | $(1.50, 0.0094)$ | $0.01 + 0.10\tau_{440}$ $0.01 + 0.03\tau_{440}$ | 1.96 |
| Desert (Bahrain-Persian Golf, 1998–2000) | $0.15$ $2.54$ | 0.42 0.61 | $(1.55, 0.0025)$ $(1.55, 0.0014)$ $(1.55, 0.0010)$ $(1.55, 0.0010)$ | $0.02 + 0.10\tau_{1020}$ $-0.02 + 0.92\tau_{1020}$ | 1.1 |
| Maritime (Lanai, Hawaii, 1995–2000) | $0.16$ $2.70$ | 0.48 0.68 | $(1.36, 0.0015)$ | $0.4\tau_{1020}$ $0.8\tau_{1020}$ | 1.4 |

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
