# Peer review of "Model Selection in Atmospheric Remote Sensing with Application to Aerosol Retrieval from DSCOVR/EPIC. Part 2: Numerical Analysis"

_remotesensing, doi:10.3390/rs12213656_

Round 1

Reviewer 1 Report

The paper studies possibilities of the different aerosol models (OPAC, MODIS, OMI, GOCART, AERONET) to correctly describe synthetic aerosol measurements from the EPIC instrument. Results of the study will be useful for the following aerosol observations by remote sensing. Only rare minor corrections are proposed. In particular:

L. 17. "selection.The algorithm" - a space before "the" is necessary.

L. 40-57. References on the statistical methods are desirable here.

L. 160. "almucantur-derived" - "almucantar" is more usual form

L. 178. "The retrieval is preformed" - probably, "The retrieval is performed"

Figure 1. There are distinctions in the caption and in the ordinate axis label: ετ(H)mean and εH(H)mean respectively. An analagous discrepancy is for Fig. 2. In L. 208 there is an inequality εH(H)max > ετ(H)mean with reference on Fig. 2.

Table 2. "The results corresponds" should be "The results correspond" (the Plural). The same correction is necessary in the caption of Fig. 5.

L. 281. A dot is visible before the first author's last name.

L. 286. The famous reference should be corrected. "Federated" is needful instead of "Federal" (!), "Setzer" instead of 'etzer'. Initials should be also checked.

L. 297. "Unequal" is correct instead of "Unegual".

Only three references from 17 are after the year 2010 (1, 2, 7). Is this a result of some crisis in the aerosol theoretical and model investigations? This feature should be commented (or supplementary new publications should be mentioned).

Author Response

Comments:
1. L. 17. "selection.The algorithm" - a space before "the" is necessary.
2. L. 40-57. References on the statistical methods are desirable here.
3. L. 160. "almucantur-derived" - "almucantar" is more usual form
4. L. 178. "The retrieval is preformed" - probably, "The retrieval is performed"
5. Figure 1. There are distinctions in the caption and in the ordinate axis
label: et(H)mean and eH(H)mean respectively. An analagous discrepancy
is for Fig. 2. In L. 208 there is an inequality eH(H)max > et(H)mean
with reference on Fig. 2.
6. Table 2. "The results corresponds" should be "The results correspond"
(the Plural). The same correction is necessary in the caption of Fig. 5.
7. L. 281. A dot is visible before the first author’s last name.
8. L. 286. The famous reference should be corrected. "Federated" is needful
instead of "Federal" (!), "Setzer" instead of ’etzer’. Initials should be also
checked.
9. L. 297. "Unequal" is correct instead of "Unegual".

Answer: We have made the suggested changes.

Comment: Only three references from 17 are after the year 2010 (1, 2, 7). Is
this a result of some crisis in the aerosol theoretical and model investigations?
This feature should be commented (or supplementary new publications should
be mentioned).

Answer: Since our retrieval method is completely new, we paid attention only
to the references that are strictly connected with our analysis, and in particular,
to the references in which the aerosol databases used in the simulations are
described.

Reviewer 2 Report

The paper of Sasi et al. describes a sensitivity analysis of an algorithm developed for the retrieval of aerosol parameters applied to synthetic EPIC measurements. A necessary step for any retrieval algorithm is the characterization of the retrieved parameters when the real atmospheric conditions are unknown. While the paper is interesting and well written, I cannot find information about the applicability of the designed algorithm. I assume that the authors explain where the algorithms could be used, i.e. is it designed only for EPIC or other sensors as well (?), in the previous part, but I believe it would also be good to mention a few things in part 2 too.

Otherwise, I only have just a few comments

Section 4:

  1. to which wavelength the AOTs correspond?   
  2. The selection of a constant albedo value (A = 0.06, which seems realistic only over water) for so different areas (i.e., industrial area, forest and dessert) could be another reason for errors. Possibly you could investigate the effect of the albedo used in the retrieval too.

Author Response

Comment: I cannot find information about the applicability of the designed
algorithm. I assume that the authors explain where the algorithms could be
used, i.e. is it designed only for EPIC or other sensors as well (?), in the
previous part, but I believe it would also be good to mention a few things in
part 2 too.

Answer: In the Introduction we specified: “In practice, the algorithm can be
used for the retrieval of (i) aerosol parameters from different spectral instruments,
and (ii) cloud parameters, when the selection of the cloud type and/or
the effective radius may play an important role.”

Comment. To which wavelength the AOTs correspond?

Answer: In the Introduction we specified the channels of the EPIC instrument
that are used in the retrieval by adding the following sentence: “Specifically,
Channels 7 and 8 in the Oxygen B-band at 680 and 687.75 nm, respectively, and
Channels 9 and 10 in the Oxygen A-band at 764 and 779.5 nm, respectively, were
used for the retrieval.”

Comment: The selection of a constant albedo value (A = 0.06, which seems
realistic only over water) for so different areas (i.e., industrial area, forest and
dessert) could be another reason for errors. Possibly you could investigate the
effect of the albedo used in the retrieval too.

Answer: The reviewer is right, the surface albedo is unrealistic. However,
we do not expect that a more realistic value of the surface albedo will change the
conclusions. The reason is that (i) the surface albedo is assumed to be known (it
is not a part of the retrieval), and (ii) both the measured and simulated spectra
are generated by a radiative transfer model using the same value of the surface
albedo. This aspect is indeed relevant when dealing with real measurements, or
when the surface albedo is also retrieved.